# Pathophysiological Effects of Autoantibodies in Autoimmune Encephalitides

**DOI:** 10.3390/cells13010015

**Published:** 2023-12-20

**Authors:** Matias Ryding, Anne With Mikkelsen, Mette Scheller Nissen, Anna Christine Nilsson, Morten Blaabjerg

**Affiliations:** 1Department of Clinical Research, University of Southern Denmark, 5000 Odense, Denmark; christine.nilsson@rsyd.dk; 2Neurobiology Research, Institute of Molecular Medicine, University of Southern Denmark, 5000 Odense, Denmark; 3Department of Clinical Immunology, Odense University Hospital, 5000 Odense, Denmark; anne.with.mikkelsen@rsyd.dk; 4Department of Neurology, Odense University Hospital, 5000 Odense, Denmark; mette.scheller.nissen2@rsyd.dk; 5Brain Research—Inter Disciplinary Guided Excellence (BRIDGE), 5000 Odense, Denmark

**Keywords:** autoimmune encephalitides, molecular mechanisms, neuroinflammation

## Abstract

The heterogeneity of autoantibody targets in autoimmune encephalitides presents a challenge for understanding cellular and humoral pathophysiology, and the development of new treatment strategies. Thus, current treatment aims at autoantibody removal and immunosuppression, and is primarily based on data generated from other autoimmune neurological diseases and expert consensus. There are many subtypes of autoimmune encephalitides, which now entails both diseases with autoantibodies targeting extracellular antigens and classical paraneoplastic syndromes with autoantibodies targeting intracellular antigens. Here, we review the current knowledge of molecular and cellular effects of autoantibodies associated with autoimmune encephalitis, and evaluate the evidence behind the proposed pathophysiological mechanisms of autoantibodies in autoimmune encephalitis.

## 1. Introduction

In recent years, recognition of autoantibody-associated neurological diseases has rapidly increased and hence the number of autoantibodies has rapidly grown. More than 20 autoantibodies associated with autoimmune encephalitides (AE) have currently been reported, which has greatly increased our understanding of central nervous system (CNS) autoimmunity, and has led to significant changes in clinical practice [1]. In AE, autoantibodies target neuronal proteins with either intracellular (classical paraneoplastic neurological syndromes) or extracellular antigens (classical AE). Pathophysiology varies for each subtype which results in different clinical phenotypes and presents a major challenge in the diagnosis of AE [2].

Autoantibodies may be directly pathogenic, the secondary consequence of tissue injury, or merely a biomarker of an autoimmune response. In general, autoantibodies towards extracellular antigens are considered to have a pathogenic role. They may exert their effect either directly on the target antigen or on the whole organ, through recruitment and activation of complement or immune cells, that initiate an attack on the tissue [3]. Autoantibodies towards intracellular antigens are in general considered to be non-pathogenic as the antigens are thought inaccessible. The pathogenic mechanisms mediated by autoantibodies are many, and except direct modulation of antigen functions, the effects depend on antibody class and subclass, which is determined by their constant domain [4]. Antibodies are one of the links between the adaptive immune system and the innate immune system and its effector mechanisms, as they do not only bind antigens, but also provide binding sites for many innate adaptor molecules or receptors. Of the five immunoglobulin isotypes (IgM, IgD, IgG, IgA, and IgE), IgG is the most abundant and often dominates the pathogenic autoimmune response [5,6]. The four subclasses of IgG (IgG1-4) mediate specific immunological processes, and may reflect important pathophysiological patterns in autoimmune diseases [6]. A high degree of sequence homology exists between the IgG subclasses, and their different immunological effector functions are due to minor amino acid differences, in particular in their hinges and upper constant CH2 domains [5]. These regions are involved in binding to the protein C1q, the recognition molecule of the classic complement cascade, and interaction with IgG-Fc receptors (FcyR) on immune cells. As a result, the effector function of the different subclasses differ in terms of immune complex formation, complement activation, and binding of FcyR-expressing cells, which may result in antibody-dependent cell cytotoxicity or cellular phagocytosis [7]. In particular, IgG1 and IgG3 mediate the crosslinking of antigens and remove them from the cell surface by internalization, activation of complement, and strong activation of FcyR [8]. IgG4 induces more subtle responses through protein blockage and sparse complement activation [8]. FcyRs have also been shown to mediate internalization of autoantibodies towards nuclear antigens upon binding and may have a role in disease pathogenesis [9,10]. Fc-receptors are widely distributed in neural tissue, and IgG uptake by neurons have been described [9,11]. The specific effector functions that are triggered by autoantibodies are also determined by the variable antigen-binding fragment (Fab), which determines antigen binding specificity and affinity [4]. But not all autoantibodies are pathogenic, and not all autoantibodies contributing to a polyclonal response are equally pathogenic. This might be due to differences between Fab-mediated diversity in epitope specificity or Fc-mediated effects.

In AE, the pathogenic roles of several autoantibodies have been described in various clinical syndromes, in particular when they bind an extracellular antigen. Many of the subtypes initially present with symptoms normally observed in neurodegenerative, neuroinflammatory, infectious or psychiatric disorders, often delaying the correct diagnosis [12,13,14]. Identification of neuronal autoantibodies is important, independent of their pathogenic potential, due to a high correlation between CSF autoantibodies and various syndromes and disease subtypes. For some autoantibodies, titers or concentrations in the CSF correlate with distinct clinical features or disease outcome [15,16]. Their detection enables a more precise diagnosis, but it is not without challenges when they are routinely screened in the clinical practice. Serum autoantibodies, especially in low titers, have been found in patients without causing symptoms associated with any specific syndrome, and their relevance might depend on the integrity of the blood–brain barrier and/or production of intrathecal autoantibodies [15,17,18,19]. To improve the diagnostic testing for autoantibodies, two different assays should be applied and both serum and CSF analyzed [20,21]. Treatment involves a combination of several immune suppressors in a first- and second-line regimen. First-line therapy includes steroids, intravenous immunoglobulins and/or plasma exchange, with escalation to second-line therapy often involving anti-CD20 therapy or cyclophosphamide [22].

The pathogenicity of autoantibodies associated with AE has been examined to a variable extent. Some have been proven to be directly pathogenic and proposed disease mechanisms have been described. However, whether all of the autoantibodies with reported pathogenicity are indeed pathogenic, and if so, through which mechanisms, has not yet been systematically evaluated. Detailed cellular disease mechanisms, besides autoantibody binding, for most subtypes are poorly understood, which hinders both the understanding of disease pathology and development of more subtype specific treatment strategies.

The importance of defining the pathogenicity of autoantibodies was already addressed in 1957, when the postulates of Witebsky et al. for defining an autoimmune disease were formulated [23]. They have since been modified and adjusted for several different types and aspects of autoimmune diseases [24,25]. Koneczny has defined aspects that should be considered when studying IgG4-specific autoimmunity, and has proposed a classification system for the evidence level of IgG4 pathogenicity in IgG4 autoimmune diseases [26].

In this review, autoantibodies towards extracellular and intracellular synaptic antigens associated with AE will be evaluated according to criteria that should be considered to define autoantibody pathogenicity. They will be divided into three categories based on different evidence levels for autoantibody pathogenicity, adapted from Koneczny’s classification system [26].

Panel 1: Criteria for autoantibody pathogenicity in autoimmune encephalitides


**
Class I evidence—Confirmed pathogenicity
**


Conclusion can be made when the following criteria have been met: -Autoantibodies targeting specific extracellular or intracellular synaptic antigens are present, and preferably identified in the affected organ-In vitro experiments have demonstrated pathogenic mechanisms for the autoantibodies, which have been validated by an independent research group

OR

-In vivo studies have reproduced the autoimmune disease either by passively transferring patient serum or purified autoantibodies, or through active immunization with the antigen. The studies have been validated by an independent research group


**
Class II evidence—Pathogenicity highly suspected
**


Conclusion can be made when the following criteria have been met:-Autoantibodies targeting specific extracellular or intracellular synaptic antigens are present, and preferably identified in the affected organ-In vitro or in vivo experiments have indicated a pathogenic mechanism for the autoantibody-Results have not yet been validated by an independent research group


**
Class III evidence—pathogenicity has not been established
**


Conclusion can be made when the following criteria have been met:-Autoantibodies targeting specific extracellular or intracellular synaptic antigens are present, and preferably identified in the affected organ-Pathogenic mechanisms are unknown, as no in vitro or in vivo studies have been reported

In the following sections, we review the characteristics of and current knowledge on potential pathogenic mechanisms of autoantibodies associated with AE, where a direct pathogenic mechanism has been described. We will classify the reported autoantibodies based on our modified criteria for autoantibody pathogenicity (see panel 1 for criteria and Table 1 for a quick overview). Clinical findings and phenotypes are summarized in Table 2, but will not be described in detail, as they have been described elsewhere [27,28,29,30].

## 2. Autoantibodies with Extracellular Targets

Extracellular antigenic proteins, such as cell surface receptors and ion channels, are readily accessible to autoantibodies, and can cause disease by altering protein function or surface abundance [10]. Since the description of anti-*N*-Methyl-D-Aspartate receptor (NMDAR) autoantibodies in 2007, a number of different surface autoantibodies associated with AE have been described [31,94,95,96]. Autoantibodies towards ionotropic receptors, such as NMDARs, often cause cross-linking and internalization of the receptor, thus reducing its surface expression. Other proposed mechanisms of actions include blockade of receptor signaling and disruption of protein–protein interaction with subsequent loss of cellular connectivity [1]. The ongoing discovery of autoantibodies not only improves diagnostic and therapeutic opportunities, but also increases our understanding of their roles in neurological diseases. In the next sections, we will describe the current knowledge on the pathogenic mechanisms of these autoantibodies. See Table 2 for an overview. Only Class I and II autoantibodies will be described in detail, as the potential pathogenic mechanisms of Class III autoantibodies are yet to be discovered.

### 2.1. Class I: Autoantibodies with Confirmed Pathogenicity

#### 2.1.1. NMDAR Autoantibodies

Patients with anti-NMDAR encephalitis are predominantly young females, but the disease can manifest in both sexes and all age groups. The clinical phenotype consists of a complex, but now classical syndrome of psychiatric symptoms, cognitive impairment, dyskinesias, seizures and autonomic instability, which can result in a catatonic state and coma [30]. The NMDAR is a ligand-gated ionotropic glutamate receptor (iGluR) and mediates excitatory neurotransmission. NMDARs are decisive for rapid regulation of neurotransmission underlying synaptic plasticity, including long-term potentiation (LTP) and long-term depression (LTD), processes important for learning and memory function [97,98,99]. The NMDARs are present in a high density within the hippocampus and the cerebral cortex, where they often work in concert with other iGluRs, α-amino-3-hydroxy-5-methyl-4-isoxazolepropionic acid receptors (AMPAR) and kainate receptors [100,101]. Functional NMDARs are heteromeric complexes consisting of two NR1 subunits in combination with two NR2 and/or NR3 subunits [98,102,103,104]. NR1 and NR3 subunits contain glycine binding sites, while glutamate binds to NR2 subunits. Most neuronal NMDARs are composed of NR1 and NR2 subunits, and NMDARs formed by these subunits requires simultaneous binding of glutamate and glycine as well as membrane depolarization to remove the magnesium block of the channel, and to allow calcium and sodium influx [105,106].

Autoantibodies towards the NR1 subunit of NMDAR is one of the most common causes of AE [107]. Autoantibodies from patients with anti-NMDAR encephalitis are predominantly of the IgG1 subclass, and thus have the potential for crosslinking, complement activation, and recruitment of inflammatory infiltrates [108]. The NMDAR autoantibodies have been shown to cause reversible internalization of the antigen in rodent hippocampal neurons as well as mouse and rat models in vitro and in vivo [31,32,34,109] (Figure 1). NMDAR autoantibodies are directed against extracellular epitopes of the NR1 subunit and cross-link receptors before they are internalized and undergo lysosomal degradation [31,32,33,34]. This leads to decreased neuronal excitability and suppression of NMDAR-dependent LTP in hippocampal neurons [35,36,37]. LTP is an important aspect of memory and in vivo studies have found that NMDAR autoantibodies impair memory, a characteristic clinical manifestation in patients with anti-NMDAR encephalitis [41,42,43,110]. Contrary to their effect on hippocampal neurons, NMDAR autoantibodies on cortical neurons seem to inhibit signaling from inhibitory to excitatory neurons, causing cortical hyperexcitability [111]. This has been hypothesized to be a possible mechanism behind NMDAR autoantibody-induced psychiatric symptoms, as NMDAR hypofunction in inhibitory cortical interneurons has been linked to schizophrenia [112].

NMDAR is not the only type of receptor affected by NMDAR autoantibodies. It has been shown that NMDAR autoantibodies decreased dopamine D1 receptor (D1R) trafficking in cultured rat hippocampal neurons [39]. Since NMDAR and D1R function as a receptor complex on the cell surface, blocking of NMDAR-D1R by the autoantibody abolishes the effect of NMDAR on D1R. It has been hypothesized that alterations in dopaminergic receptors could play an important part in the development of the psychotic features seen in anti-NMDAR encephalitis. Similarly, an in vivo murine study showed decreased surface D1R and increased surface dopamine D2 receptor (D2R) in mice exposed to patient NMDAR IgG [40]. Interestingly, it has been reported that NMDAR autoantibodies not only affect neurons, but also oligodendrocytes, as patient NMDAR autoantibodies decreased oligodendrocyte NMDAR response to glutamate and their glucose transporter 1 (GLUT1) expression (Figure 1) [38]. Activation of NMDARs on oligodendrocytes has previously been shown to be important for metabolic regulation of oligodendrocytes through recruitment of GLUT1 to the cell membrane, which is also important for the myelination process [113]. The same study showed that loss of oligodendrocyte NMDARs leads to inflammation and axonal degeneration, revealing a potential novel disease pathway of anti-NMDAR encephalitis [113].

As one of the most well-studied subtypes of AE, it is also the type in which novel treatment strategies have been explored the most. The autoantibody-mediated internalization of NMDAR has been shown to be blocked by activation of Ephrin-B2 receptors (EPHB2R), possibly through increased interaction between NMDAR and EPHB2R which is believed to stabilize NMDARs in the synapse, thus considered as a possible therapeutic target [34,35]. It has also been found that pharmacological allosteric modulation of NMDAR can prevent an autoantibody-induced reduction in NMDAR-surface expression and functional autoantibody-mediated effects such as memory loss [114]. Recently, it was found that treatment with pregnenolone sulphate, a steroid that increases trafficking of NMDAR to the cell membrane, could partially subdue seizures caused by patient NMDAR autoantibodies in an in vitro rat model [115]. However, this study did not show that this rescue was caused by increased surface expression of NMDAR.

#### 2.1.2. Leucine Rich Glioma-Inactivated 1 (LGI1) Autoantibodies

In 2001, autoantibodies against voltage-gated potassium channels (VGKC) were found in patients with limbic encephalitis [116]. It was later discovered that the actual targets were not the potassium channel subunits of the *Shaker*-family (Kv1.1, Kv1.2, or Kv1.6), but most often the associated proteins Contactin-associated protein-like 2 (CASPR2) and LGI1 [52,117,118,119]. Patients with anti-LGI1 encephalitis are typically middle-aged and present with limbic encephalitis comprising prominent amnesia and epileptic seizures. A predominant and pathognomonic feature are the so-called facio-brachial dystonic seizures [30]. LGI1 is a scaffolding protein important for AMPAR and VGKC function. The AMPARs mediate fast excitatory glutamatergic neurotransmission and synaptic plasticity, critical for normal brain function. Similarly, the VGKCs also play a pivotal role in neurotransmitter release as well as several other cellular processes, including the modulation of neuronal excitability and induction of action potentials [120,121,122]. LGI1 regulates the neuronal network by modulating neuronal excitability through the Kv1.1. potassium channel of the VGKC, and is essential for the localization of the Kv1 and Kv1.2 subunit complexes to the synaptic terminals. At the postsynaptic terminal, LGI1 regulates the synaptic transmission by acting on the expression of AMPARs. These mechanisms are mediated by binding of LGI1 to presynaptic Disintegrin, metalloproteinase domain-containing protein 23 (ADAM23), and postsynaptic ADAM22 [121,123].

Most anti-LGI1 encephalitis patients harbor autoantibodies of the IgG4 subclass, although IgG1 autoantibodies may also be present. IgG4 autoantibodies are more subtle mediators of effector mechanisms, but still able to block receptor function and protein–protein interaction. LGI1 autoantibodies interfere with LGI1 binding to ADAM22/ADAM23, thereby removing LGI1 from its place of action (Figure 2 “anti-LGI1”) [46,47,48]. This effect has been shown to be mediated by autoantibodies targeting the epitempin repeat domain of LGI1, but most LGI-1 encephalitis patients harbor autoantibodies against both the epitempin repeat and the leucine-rich repeat domain of LGI1 [46,49]. One paper has shown that patient leucine-rich repeat domain-specific monoclonal IgG4 autoantibodies recognize LGI1 bound to ADAM22/ADAM23, and induce internalization of the complex in both human embryonic kidney 293T (HEK293T) cells and live hippocampal neurons [49]. IgG4 binding is not commonly believed to be able to cause internalization, and this finding requires further validation. Binding interference has been described as an effect of exposure to polyclonal patient’s LGI1 autoantibodies, but it is possible that interference and internalization occur simultaneously in anti-LGI1 encephalitis patients. The loss of LGI1-ADAM22/ADAM23 complexes on the cell surface leads to reduced AMPAR and VGKC clusters [46,47]. Consequently, AMPA-current dependent LTP is impaired, and this is thought to be the direct cause of memory impairment seen in patients with anti-LGI1 encephalitis [47,49]. The loss of excitatory AMPAR clusters seems to be negligible in relation to neuron excitability, when compared to the loss of inhibitory VGKC clusters. Patch clamp analysis of mouse and rat neurons treated with LGI1 autoantibodies show increased excitability measured as increased spike rate and amplitude [47,48,50]. This has also been shown through silencing of LGI1 with small-hairpin RNA [124]. Interestingly, a paper published in 2022 showed that only monoclonal autoantibodies against the leucine rich domain of LGI1, and not those directed against the epitempin repeat domain, caused neuronal hyperexcitability, an effect facilitated at least partially through inhibition of the Kv1 potassium channel [125].

In a feline in vivo model of anti-LGI1 encephalitis, breakdown of tight junctions in the blood–brain barrier (BBB) was observed, specifically in the limbic regions, the areas of the brain with the highest concentration of LGI1 and thus predominantly affected by LGI1 autoantibodies [51,126]. This could potentially explain why autoantibodies against LGI1, which is distributed across the entire brain, target mainly the limbic regions and cause limbic encephalitis [51].

#### 2.1.3. CASPR2 Autoantibodies

Patients with CASPR2 autoimmunity are predominantly male, middle-aged, and may present both/or either CNS and peripheral nervous system (PNS) symptoms, such as neuropathy, neuromyotonia, neuropathic pain, Morvan syndrome, or limbic encephalitis [30]. CASPR2 is a transmembrane cell adhesion protein of the neurexin family that regulates the localization of Kv1.1 and Kv1.2 VGKCs at the juxtaparanodal region of myelinated axons, important for proper electrical function [127,128]. Apart from its presence in the node of Ranvier in the CNS and PNS, CASPR2 is also located in the limbic system, basal ganglia, and the motor and sensation areas of the brain [129]. Similar to LGI1 autoantibodies, CASPR2 autoantibodies are mainly of the IgG4 subclass and a minor part are of the IgG1 subclass. CASPR2 autoantibodies interfere with antigen interactions, and disrupt the interaction between CASPR2 and the scaffolding protein contactin-2 (Figure 2 “Anti-CASPR2”) [53]. The immediate cause of this is highly disputed. While some studies have reported interference with CASPR2 cluster formation or increased CASPR2 cell surface expression, others report no change in CASPR2 expression or decreased synaptic CASPR2 through internalization [53,54,55,57,130]. Whether this inconsistency is due to CASPR2 autoantibodies of different subclasses, the heterogeneity of specific autoantibody mechanisms or differences in study design is unknown, but it warrants further investigation.

The disturbed CASPR2-Contactin-2 connection leads to increased neuronal excitability through increased K_V_1.2 surface expression [50,54,56]. One study also observed the internalization of AMPAR clusters resulting in decreased AMPAR-mediated currents and LTP [130]. In 2022, this was supported by findings in mice exposed to CASPR2 patient autoantibodies who exhibited memory impairment caused by CASPR2 internalization [57]. In an in vivo mouse model, CASPR2 autoantibodies altered astrocyte and microglia morphology and increased complement C3 expression consistent with glial activation [55].

#### 2.1.4. AMPAR Autoantibodies

Patients suffering from anti-AMPAR encephalitis are classically middle-aged women presenting with limbic encephalitis, epileptic seizures, diffuse encephalopathy or isolated psychosis [30]. AMPARs are widely distributed in the CNS and are members of the iGluR family. Upon glutamate binding, the AMPAR opens for sodium and potassium influx, mediating fast depolarization and thus, excitatory glutamatergic neurotransmission [120]. They are enriched in the postsynaptic membrane on dendritic spines, and are highly dynamic. Changes in AMPAR number at the synapse, subunit composition, phosphorylation state, and several accessory proteins can all modify the efficacy of synaptic transmission [120,131]. The AMPARs can consist of four different subunits: GluA1-4, and autoantibodies against GluA1 and GluA2 have been found in AE patients [58,61]. The pathological effect of GluA1 and GluA2 autoantibodies is similar to that of NMDAR autoantibodies, with a decrease in GluA1 and GluA2 containing AMPAR clusters through internalization and lysosomal degradation (Figure 2 “Anti-AMPAR”) [58,59,60]. Studies on primary rat neurons and in vivo mouse models have shown that the loss of excitatory AMPAR clusters decrease AMPAR mediated currents, impair LTP and induce memory impairment [59,60,61]. The IgG subtypes of autoantibodies against GluA1 and GluA2 have not been determined, but several articles state that it is of the IgG1 subtype, possibly due to their ability to internalize their antigen [21,132].

GluA3 autoantibodies have been found in patients with Rasmussen’s encephalitis and can be found in almost a fourth of patients fulfilling the criteria for frontotemporal dementia [133,134]. GluA3 autoantibodies from frontotemporal dementia patients reduce synaptic GluA3-containing AMPARs and the dendritic spine density in rat hippocampal neurons [133,134]. The same research group observed increased tau levels in neurons derived from human induced pluripotent stem cells exposed to patient GluA3 autoantibodies [134]. As tau deposits are commonly found in frontotemporal dementia and other tauopathies, GluA3 autoantibodies may be an interesting link between neuroimmunology and neurodegeneration.

#### 2.1.5. Aminobutyric Acid (GABA) Type A Receptor Autoantibodies

Patients with anti-GABA type A receptor (GABA_A_R) encephalitis present a rapidly progressive encephalopathy, with prominent seizures and status epilepticus [30]. GABA_A_Rs are ligand-gated ionotropic chloride channels and ubiquitously expressed throughout the CNS. GABA_A_Rs are the main mediators of fast inhibitory neurotransmission in the CNS and due to their widespread localization they play a pivotal role in many neurological functions such as cognition, behavior, and consciousness [135]. Most GABA_A_Rs assemble as heteropentamers and consist of two α subunits, two β-subunits and a single γ subunit [136]. GABA_A_R autoantibodies are generally of the IgG1 subclass, and have been found to target several different sections of the receptor, including the α1, β3 and γ2 subunit [65,137]. GABA_A_R autoantibodies have been reported to reduce the expression of synaptic GABA_A_R and consequently reduce inhibitory postsynaptic currents in vitro, but the molecular mechanisms by which the autoantibodies mediate their effect on the antigen is not well known, and conflicting results have been reported (Figure 3 “Anti- GABA_A_R”) [63,64]. Unaltered surface expression of GABA_A_R clusters as well as reduced surface expression of GABA_A_R clusters have been reported when exposing rat primary neurons to autoantibodies from anti-GABA_A_R encephalitis patients [63,64,65,68]. One study reported, that even though total GABA_A_R clusters were unaltered, a specific reduction in GABA_A_R in synaptic clusters occurred [63]. Others report reduction in both total surface GABA_A_R amount and GABA_A_R in synaptic clusters [64,65].

Until recently, only a single study on the subsequent effects of GABA_A_R autoantibodies had been published, which reported decreased amplitude of inhibitory postsynaptic currents in rat hippocampal neurons [64]. This was later confirmed in hippocampal mouse neurons and in mouse brain slices exposed to GABA_A_R autoantibodies [67,68,138]. Others have reported, that the inhibitory postsynaptic signaling was reduced in both hippocampal CA1 and CA3 pyramidal neurons, which was linked to increased excitability in hippocampal CA1 pyramidal neurons [138]. The first in vivo study of GABA_A_R autoantibodies was published in 2021, and reported epileptic seizures in mice when exposed to monoclonal IgG1 patient autoantibodies [68]. Further structural analysis of GABA_A_R IgG1 autoantibodies from a single patient has revealed two distinct mechanisms of receptor inhibition and pathology: direct antagonizing of GABA binding, or blocking allosteric modulation of the GABA_A_R [66]. These findings suggest that IgG1 antibody pathogenicity can be mediated by other mechanisms than target receptor cross-linking and internalization, and that autoantibodies from one individual can bind to distinct subunits to interfere with receptor function.

#### 2.1.6. Dipeptidyl-Peptidase-like Protein-6 (DPPX) Autoantibodies

Patients suffering from anti-DPPX encephalitis are often adults of older age and may present with a rapidly progressing encephalopathy, prominent gastrointestinal symptoms and severe weight loss or progressive encephalomyelitis with rigidity and myoclonus (PERM) [30]. DPPX is a cell surface auxiliary subunit of the voltage dependent Kv4.2 potassium channels with distinct properties compared to the Kv1 family, which includes LGI1 and CASPR2 [139]. The Kv4.2 potassium channels belong to the *Shal*-type (Kv4) protein family, and are responsible for transient, inhibitory currents in the CNS and PNS. These currents regulate somatodendritic signal integration, regulation of dendritic back-propagation of action potentials into neuronal dendrites, and synaptic plasticity [140,141]. DPPX associates with the pore-forming subunit of the channels, facilitate intracellular trafficking to the plasma membrane, and hence increases the surface expression of Kv4 channels [142]. DPPX is predominantly expressed in the hippocampus, the cerebellum, and in the myenteric plexus [139].

The DPPX autoantibodies are predominantly IgG1 and IgG4, but their individual contributions to the pathogenic effect are unknown [69]. To date, two studies on the molecular and cellular effects of DPPX autoantibodies have been published. While both studies found that DPPX autoantibodies decrease the surface expression of both DPPX and Kv4.2 on hippocampal neurons, only one found this decrease reversible after autoantibody removal (Figure 4 “Anti- DPPX”) [69,70]. The exact mechanism causing the decrease in DPPX/Kv4.2 is yet to be elucidated. The effect of DPPX autoantibodies on the electrophysiological properties of CNS neurons have not yet been examined. Knockout of DPPX has been shown to cause hyperexcitability of hippocampal neurons, and patients with anti-DPPX encephalitis often have symptoms of CNS hyperexcitability [69,143]. Autoantibody-mediated downregulation of DPPX and Kv4.2 is therefore anticipated to cause CNS hyperexcitability, but this has yet to be clarified.

DPPX autoantibodies also have pronounced effects on the activity of enteric neurons.

Patient derived DPPX autoantibodies increased the excitability and action potential frequency of guinea pig and human enteric neurons, and may be the reason why the majority of patients with anti-DPPX encephalitis have prominent gastrointestinal symptoms, including diarrhea [70]. The hyperexcitability of enteric neurons was induced rapidly upon binding of DPPX autoantibodies to Kv4.2 potassium channels, why it has been hypothesized to be caused by modulation of the electrophysiological properties of the complex rather than internalization of DPPX or Kv4.2 [70].

#### 2.1.7. IgLON5 Autoantibodies

Patients with IgLON5 disease present an insidious onset with different clinical phenotypes and symptoms such as non-REM parasomnia, stridor, obstructive sleep apnea, bulbar dysfunction, cognitive decline, gait instability, dystonia or dyskinesia and PNS symptoms [30]. The clinical presentation may resemble that of other neurodegenerative diseases, such as atypical Parkinson, Huntington’s disease, or motor neuron disease. IgLON5 is a member of the IgLON protein family, consisting of adhesion molecules involved in synaptogenesis, but the specific function of IgLON5 is still unknown [144]. IgLON5 is most abundantly expressed in the nervous system, but has also been found in lower concentrations in other organs. Patients with autoantibodies against IgLON5 were first described in 2014 [77]. Autoantibodies are predominantly of the IgG4 subclass, but many patients display both IgG4 and a smaller amount of IgG1 in serum [145]. In 2016, it was reported that exposure of rat hippocampal neurons to the IgG1 fraction of patient anti-IgLON5 autoantibodies resulted in internalization of cell surface IgLON5 clusters (Figure 4 “Anti- IgLON5”) [72]. In contrast to the other AE autoantibodies, this was shown to be irreversible. As expected, the IgG4 fraction from the same patient did not cause a reduction in IgLON5 surface clusters.

Anti-IgLON5 autoantibodies have also been reported to cause dystrophic neurites and axonal swelling in rat hippocampal neurons, indicating that the autoantibodies have a direct neurodegenerative effect [75]. We have recently expanded these findings using human induced pluripotent stem cell derived neurons exposed to anti-IgLON5 autoantibodies [73]. Neurons exposed to patient anti-IgLON5 autoantibodies had reduced post- and presynaptic clusters, decreased spike rates, and accumulation of phosphorylated-tau, a general marker of neurodegeneration often found in the brains of IgLON5 disease patients post-mortem [73,146]. Finally, neuronal cultures exposed to anti-IgLON5 autoantibodies for three weeks or more exhibited increased cell death, demonstrating that the IgLON5 autoantibodies in these patients are the cause of neurodegeneration. Recently, reduced spike rate and synapse number were confirmed as an effect of IgLON5 autoantibodies in an in vivo study in mice with passive transfer of patient autoantibodies (7–10 days of transfer) [74]. Seven days after the passive transfer was ended, no increase in cell death was detectable, however, 30 days after, mice exhibited increased neuronal cell death. This indicates, that the effects of IgLON5 autoantibody binding are irreversible, and that perhaps removal of patient autoantibodies as a treatment strategy is inadequate. In another study, mice exposed to patient IgLON5 autoantibodies displayed cognitive abnormalities and increased density of microglia and astrocytes in the hippocampus, suggesting an inflammatory response. Recently, findings in in vivo wild type mice and mice expressing human tau protein infused with patient IgLON5 autoantibodies were published. This study confirmed p-tau deposition (not quantified), and minor changes in breathing and behavior in female, but not male mice [76]. Perhaps differences in autoantibody epitope recognition or specifically anti-IgLON5 autoantibody concentration is the explanation for this.

#### 2.1.8. Glycine Receptor (GlyR) Autoantibodies

The clinical phenotype of patients with GlyR autoimmunity is dominated by rigidity and stiffness, and patients present with syndromes such as PERM or Stiff Person Spectrum disorders (SPSD) [30]. GlyRs are ligand-gated chloride channels and mediate fast inhibitory neurotransmission in the brainstem and spinal cord, where they are primarily involved in motor control and pain perception [147,148]. GlyR autoantibodies towards GlyRα1 were first reported in 2008, and since then, further patients with GlyR autoantibodies have been described [149]. GlyR autoantibodies are predominantly of the IgG1 and IgG3 subclass and activate complement on the cell surface of live GlyRα1 expressing HEK cells [78].

GlyR autoantibodies have been reported to cause internalization and lysosomal degradation of GlyRs on HEK293T cells transfected with GlyR, a mechanism similar to what is seen with NMDAR and AMPAR autoantibodies (Figure 4 “Anti- GlyR”) [78]. Reduced expression of GlyR has also been demonstrated in zebra fish larvae. In vivo transfer of GlyR autoantibodies reduced the number of GlyrR clusters in the spinal cord and reduced the animal’s response to tactile stimulation [150]. In this study, GlyR autoantibodies also reduced the cellular response to glycine when performing a patch clamp experiment on HEK cells transfected with GlyR. An in vivo study on the behavioral effects reported increased anxiety, but no motor effects, in a murine disease model [80]. Some patients with anti-GlyR encephalitis suffer from PERM, thus motor effects could be expected as an effect of GlyR autoantibodies. Recently, it was also reported that patient GlyR autoantibodies decreased glycinergic transmission in rat motor neurons [79]. Moreover, Fab fragments generated from GlyR autoantibodies had the same effect, and since Fab fragments are not able to cause cross-linking and internalization, it has also been suggested that that GlyR autoantibodies are able to directly inhibit GlyR [79].

### 2.2. Class II: Autoantibodies with Highly Suspected Pathogenicity

#### 2.2.1. GABA Type B Receptor (GABA_B_R) Autoantibodies

Patients suffering from anti-GABA_B_R encephalitis are predominantly male and present with limbic encephalitis and prominent seizures [30]. GABA_B_Rs are G protein-coupled receptors, expressed widely in the CNS and mediate slow and prolonged inhibitory actions, including inhibition of neurotransmitter release and modulation of neuronal excitability [151]. The majority of GABA_B_Rs are heterodimers of GABA_B1_ and GABA_B2_ subunits and heterodimerization is required for the cell surface expression and signaling by GABA_B_R [152]. GABA_B_R autoantibodies are mainly of the IgG1 subclass and targets the extracellular domain of the GABA_B1_ subunit, where the recognition site for GABA is located [82]. The pathological mechanisms of GABA_B_R autoantibodies are poorly understood and the current knowledge is disputed. In one study, application of patient autoantibodies to primary hippocampal rat brain slices, did not alter the neuronal cell-surface expression of GABA_B_R, but may have had a direct effect on receptor function, leading to a reduction in spike rate in layer III pyramidal cells of the medial entorhinal cortex which the authors hypothesized could result in reduced inhibition and thereby induce hyperexcitability in the medial entorhinal cortex (Figure 3 “Anti- GABA_B_R”) [153]. A more recent study from 2019 cast doubt on this hypothesis as the signaling patterns of murine brain slice cultures were unaltered upon treatment with GABA_B_R autoantibodies [50].

#### 2.2.2. Metabotropic Glutamate Receptor 1 (mGluR1) Autoantibodies

The clinical phenotype of anti-mGluR1 encephalitis is dominated by subacute onset of cerebellar ataxia. mGluR1 is a G protein-coupled cell surface receptor and belongs to the group I family of mGluRs. These receptors are distributed in various regions the CNS, with a prominent expression in especially the Purkinje cells of the cerebellum, where they are involved in LTD of Purkinje cell-parallel fiber synapses, which are important for cerebellar coordination and motor learning [154,155]. The first description of autoantibodies against mGluR1 was described two decades ago from two patients with cerebellar ataxia [85]. The autoantibodies from these patients could block the production of inositol phosphates, a downstream product of mGluR1 signaling, in Chinese hamster ovary cells. When the autoantibodies were injected into the cerebellar subarachnoid space of mice, it caused severe but reversible ataxia.

The same research group later reported that cerebellar mice slice cultures exposed to these patient autoantibodies had a reduced response to mGluR1 agonists measured as membrane potential changes [84]. In cultured mouse Purkinje cells, patient autoantibodies reduced LTD. An in vivo experiment using osmotic pumps to transfer patient autoantibodies to mice showed decreased compensatory eye movements [84]. Recently another research group reported that rat hippocampal neurons exposed to mGluR1 autoantibodies of mainly IgG1 subclass exhibit a decreased number of mGluR1 clusters in cultured neurons [83]. Although three publications have described the pathological mechanisms of mGluR1 autoantibodies, the studies were performed by only two independent research groups and attempted replication of previous findings were not reported. Therefore, further studies are required to validate these findings and to characterize the potential pathogenic mechanisms mediated by the autoantibodies.

#### 2.2.3. Metabotropic Glutamate Receptor 5 (mGluR5) Autoantibodies

Anti-mGluR5 encephalitis can be seen in both children and adults, and it mostly presents with subacute memory loss and/or psychosis [30]. mGluR5 constitute together with mGluR1 the group I family of mGluRs, and both are particularly important for synaptic plasticity including LTP and LTD. In particular, mGluR5 is important for the associative strengthening of neuronal connections during learning and memory, and is highly expressed in regions of the brain implicated in these mechanisms, including the hippocampus and lateral nucleus of amygdala [156]. mGluR5 receptors are also expressed in the cerebellum, but to a lesser degree [157]. The implication of mGluR5 to synaptic plasticity and learning may be regulated by modulation of NMDARs [158,159]. In addition, mGluR5 may also play a role in contextual fear conditioning. It has been shown that mGluR5 expression increases in hippocampus during contextual fear conditioning and is impaired in mice deficient for mGluR5, and that acquisition of the fear response can be disrupted by mGluR5 antagonists [156,160,161,162]. Activation of mGluR5 by glutamate primarily enhances postsynaptic excitation, but it can also act on presynapses where it can enhance neurotransmitter release [163]. The first cases of patients with autoantibodies against mGluR5 were described in 2011 [164]. However, nothing was known about the pathological mechanisms of the autoantibodies until 2018, when it was found that patient mGluR5 autoantibodies, mainly of the IgG1 subclass, decreased the amount of mGluR5 clusters on rat hippocampal neurons [86]. In 2019, it was further reported that the patient’s autoantibodies caused anxiety and progressive memory loss in mice [165]. The two publications have been authored by the same group and none of the pathological mechanisms have been validated by an independent research group.

#### 2.2.4. Neurexin-3α Autoantibodies

Patients suffering from anti-neurexin-3α encephalitis are predominantly female and experience confusion, seizures and decreased consciousness [30]. The clinical syndrome resembles that of anti-NMDAR encephalitis. The target antigen of neurexin-3α autoantibodies is a cell adhesion molecule involved in synapse formation and function, but the exact role of neurexin-3α at the synapse is still unknown [166]. Interestingly, α-neurexins have been demonstrated to be important for localization and function of synaptic Ca^2+^ channels. These findings suggest that neurexin-3α are required for Ca^2+^ triggered presynaptic neurotransmitter release at synapses and thus important for synaptic transmission [167]. In 2016 the first publication reporting five patients with autoantibodies against neurexin-3α was published, and is still the only paper to describe pathogenic effects of the autoantibodies [87]. The IgG isotype was investigated in four of the five patients, and were only IgG1 or IgG1 with a minor component of IgG4. Primary cultures of rat hippocampal embryonic neurons exposed to patient autoantibodies for 10 days showed reduced expression of neurexin-3α as well as decreased number of synapses. In mature rat neurons, the expression of neurexin-3α was decreased, but the total number of synapses were unaffected after 48 h of autoantibody exposure. These findings suggest that neurexin-3α affects synapse development, but further studies are necessary to evaluate whether or not neurexin-3α have an effect on the maintenance of mature neurons and may affect pre- and postsynaptic functions.

#### 2.2.5. Kainate Receptor Subunit 2 (GluK2) Autoantibodies

Anti-GluK2 encephalitis may present with headache, nausea and vomiting and progress to acute cerebellitis with the risk of hydrocephalus [88]. Almost all patients present ataxia, but some do not present with acute progressive cerebellitis. Instead they present with confusion, altered behavior and cognitive decline [88]. The kainate receptor belongs to the family of iGluRs alongside AMPARs and NMDARs. Kainate receptors are widely distributed and highly expressed pre- and post-synaptic throughout the CNS where they modulate neuronal circuits and neural activity by modulation of glutamatergic and GABAergic signal transmission [168,169,170]. This modulation of neuronal excitability is mediated by either their ionotropic activity or through interaction with metabotropic signaling. Presynaptic kainate receptors regulate neurotransmitter release, whereas postsynaptic KARs mediate modulation of excitatory neurotransmission [168,169,171].

GluK2 can form homomeric receptors and their function has been extensively investigated in hippocampal neurons, but are expressed in many other CNS structures, including the cerebellum [172,173,174]. Presynaptic Gluk2 are involved in mossy fiber short- and long-term potentiation in the hippocampus and thus have the ability to modulate synaptic plasticity properties [175,176]. Postsynaptic GluK2 kainate receptors in mature hippocampal pyramidal neurons signals primarily via metabotropic pathways and regulates synaptic excitability through inhibition of the slow afterhyperpolarization current [177]. Autoantibodies of the IgG1 subclass against the glutamate kainate receptor has been described in a case series of eight patients and the GluK2 was identified as the main antigen target [88]. When exposing primary hippocampal rat neurons to GluK2 autoantibodies associated with AE, the levels of cell-surface and synaptic GluK2 were decreased in a reversible manner [88]. The patient’s sera decreased GluK2-mediated currents in GluK2-expressing HEK293 likely due to GluK2 internalization and thus resembles mechanisms similar to NMDAR and AMPAR autoantibodies. To date, the study of Landa et al. 2021 is still the only publication to show pathogenic effects of GLUK2 autoantibodies which merits further research [88].

#### 2.2.6. Ca_V_α2δ Autoantibodies

In 2021 autoantibodies of the IgG2 subclass targeting an auxiliary subunit of presynaptic voltage-gated calcium channel (VGCC or Ca_V_), α-2/δ subunit (Ca_V_α2δ) were found in two patients by screening serum- and CSF samples from seronegative AE patients [89]. The two patients presented with subacute cognitive decline, altered behavior, seizures and decreased consciousness. One, an adolescent girl, after presumed viral meningitis. The other, an older male, concomitant with a neuroendocrine tumor. VGCCs are the main mediators of calcium entry into neurons in response to membrane depolarization and couples neuronal excitability and neuronal neurotransmitter release. The molecular diversity of these calcium channels are large and are composed of a pore-forming Ca_V_α1 and auxiliary Ca_V_α2δ and Ca_v_β subunits [178]. Functionally, Ca_V_α2δ increase the expression of the VGCC at presynaptic terminals and enhance tight coupling of Ca^2+^ channels to exocytosis and subsequently neurotransmitter release [179]. In cultured rat hippocampal neurons, binding of autoantibodies to the Ca_V_α2δ subunit of VGCC weakened the coupling of VGCCs and exocytosis, suggesting that the autoantibodies inhibit neurotransmitter release. Whole cell-recording of cultured rat hippocampal neurons showed that autoantibodies impaired neuronal excitatory and inhibitory postsynaptic current frequencies, potentially by interfering with the tight coupling of Ca^2+^ channels and Ca^2+^ sensors [89]. These findings needs to be validated, and warrant further research to evaluate the pathogenicity of IgG2 autoantibodies against Ca_V_α2δ.

### 2.3. Class III: Autoantibodies without Established Pathogenicity

Autoantibodies against D2R, metabotropic glutamate receptor 2 (mGluR2), Seizure Related 6 Homolog Like 2 (SEZ6L2) and Glutamate receptor δ2 (GluRδ2) have been associated with rare cases of AE [91,92,93,132,180,181]. It cannot be excluded that these autoantibodies have direct pathogenic effects, but the first report is yet to be published addressing molecular and cellular effects. Please see Table 2 for a short description of the clinical phenotypes of AE with these types of autoantibodies.

## 3. Autoantibodies with Intracellular Synaptic Targets

The following sections describe autoantibodies with intracellular targets involved in synaptic transmission. Although intracellular, the antigens have been hypothesized to be exposed to the extracellular space temporarily during exocytosis of neurotransmitter-containing vesicles [182]. Another theory states that autoantibodies may be internalized into synapses during endocytosis of vesicular components followed by fusion of autoantibody-containing vesicles with antigen-containing vesicles [182]. Thus, providing possible mechanisms of how these autoantibodies may come into direct contact with their antigens.

### 3.1. Class I: Autoantibodies with Confirmed Pathogenicity

#### Glutamic Acid Decarboxylase 65 (GAD65) Autoantibodies

Glutamic acid decarboxylase (GAD) is the rate-limiting enzyme of GABA synthesis that catalyzes alpha-decarboxylation of L-glutamate into GABA [183]. Two GAD isoforms exist: GAD65 and GAD67. The larger isoform, GAD67, is located in the cytoplasm whereas the smaller, GAD65, is predominantly located in the nerve terminals of GABAergic neurons, anchored to the membrane of synaptic vesicles and is also involved in the packaging, and release of GABA [184,185,186,187]. GAD autoantibodies are predominantly of the IgG1 subclass and can be associated with several neurological disorders, including SPSD, cerebellar ataxia, epilepsy and more rarely limbic encephalitis [188,189,190,191,192,193,194]. Besides neurological immune-mediated disorders, GAD65 autoantibodies are also present in type 1 diabetes mellitus (T1DM) and is an important biomarker for disease progression [195].

GAD67 autoantibodies are almost always accompanied by GAD65 autoantibodies in patients with neurological syndromes and are not believed to induce neurological symptoms on their own, wherefore they are not considered to be of clinical importance [196]. Although, GAD65 autoantibodies have been extensively studied for their involvement in neurological disorders, the potential access route to the intracellular antigen is currently unknown. In vitro studies have demonstrated internalization of human monoclonal GAD65 autoantibodies in a rat mesencephalic cell line or into neurons surrounding the area of injection into rat cerebral cortex in vivo [197,198]. However, another group did not observe internalization of GAD65 when administering autoantibodies obtained from serum of patients with high-titer GAD65 autoantibodies to cultures of live rat hippocampal neurons [193]. Alternatively, GAD65 autoantibodies may interact with GAD65 during exocytosis, given that the antigen is temporarily exposed [199,200]. This has only been hypothesized, and none of these studies have shown that the autoantibodies react with either temporarily extracellular exposed or intracellular GAD65.

The pathological mechanism by which GAD65 autoantibodies might interfere with its target antigen is yet to be unveiled. In vitro studies have shown that GAD65 autoantibodies from patients with SPSD in most cases inhibit the enzymatic activity of GAD65 and reduce GABA synthesis in opposition to GAD65 autoantibodies from patients with T1DM, which have no such effect [201,202]. Several studies have characterized differences in GAD65 autoantibodies titer and epitope specificity present in neurological syndromes compared to GAD65 autoantibodies present in T1DM. The level of circulating GAD65 autoantibodies in neurologically ill patients often exceed those for patients with T1DM by 10 to 100-fold and recognize linear epitopes, whereas GAD65 autoantibodies in T1DM primarily react with conformational epitopes [189,190,201,203,204,205,206]. In this context, it has been suggested that the distinct neurological impairments caused by GAD65 autoantibodies vary according to epitope specificity rather than autoantibody levels. Studies with human monoclonal GAD65 autoantibodies with diverse epitope specificities or GAD65 autoantibodies from patients with different neurological syndromes have demonstrated epitope specificity of GAD65, however, these observations have been challenged as others failed to identify syndrome-specific epitope autoantibodies [193,197,202,207,208,209]. In support of an epitope-dependent manner of action by GAD65 autoantibodies, it has been demonstrated in vitro that GAD65 autoantibodies in cerebellar ataxia interfere with GABA release, either by inhibiting the packaging of GABA into vesicles and/or by impeding with the shuttling of vesicles [207,208,210], whereas GAD65 autoantibodies in SPSD block the synthesis of GABA [201,205,208].

To evaluate the effect of GAD65 autoantibodies on cerebellar GABAergic transmission, studies with electrophysiological recordings have also been performed. The effect has primarily been studied in patch-clamp experiments in rat cerebellar slices and cultured rat hippocampal neurons, when treated with CSF from patients with GAD65-autoantibody associated cerebellar ataxia and epilepsy. While some studies did not observe disturbances in the GABAergic transmission, several others found that GAD65 autoantibodies caused suppression of GABAergic transmission and consequently interfered with neuronal inhibition [211,212,213,214,215,216,217]. The CSF from patients with cerebellar ataxia acted on nerve terminals of GABAergic interneurons, impairing the inhibitory synaptic transmission mediated by GABA onto Purkinje cells, and on parallel fibers, attenuating the release of excitatory glutamate onto Purkinje cells. This dual impairment of inhibitory synaptic transmission resulted in hyperexcitability of cerebellar Purkinje cells and corresponds well to the clinical presentation with ataxia observed in patients [212].

Studies in animal models have shown that injections of GAD65 autoantibodies induced clinical symptoms and neurophysiological changes similar to those seen in patients. These studies have reported increased excitability of the spinal cord, increased neuronal synaptic function, stiffness-like motor deficits, behavioral changes and impaired cognitive functions [197,198,207,218,219]. But these experiments failed to demonstrate whether the observed changes were directly caused by dysregulation of GAD65 by the autoantibodies. Also, mice immunized with GAD65 have been shown to induce autoantibody production, but not to develop neurological symptoms, despite high titers of the autoantibody in serum [220,221]. 

While mechanistic details on how the autoantibodies may reach their antigen still needs to be uncovered, and their diverse effects in neurological disorders explained, various studies support a pathogenic role of GAD65 autoantibodies wherefore they are generally believed to have pathogenic potential.

### 3.2. Class II: Autoantibodies without Established Pathogenicity

#### 3.2.1. Amphiphysin Autoantibodies

Autoantibodies against amphiphysin were first discovered in 1992 in patients with SPSD, and are often associated with small cell lung cancer [222,223,224]. Amphiphysins belong to the Bin-Amphiphysin-Rvsp (BAR) domain family of proteins that are involved in endocytosis of synaptic vesicles through interactions with protein in the clathrin-coated complex, such as clathrin, synaptojanin, and dynamin [225,226]. Amphiphysin I is highly expressed in presynaptic nerve terminals whereas Amphiphysin II is enriched in axon initial segments and nodes of Ranvier [225]. Amphiphysin I may form a homodimer or a heterodimer with Amphiphysin II, which is believed to increase the stability of Amphiphysin II [227].

Passive transfer of a patient’s amphiphysin autoantibodies to rats with a permeable BBB induced spasms resembling SPSD, indicating that the autoantibodies had a direct pathological effect [228]. The same research group have also described a direct effect of amphiphysin autoantibodies on cultured rat embryonic motor neurons: amphiphysin autoantibodies blocked GABA-induced intracellular Ca^2+^-rise and reduced surface expression of NKCC1, a Na^+^/K^+^/2Cl^2^—cotransporter important for motor neuron homeostasis [229]. Furthermore, infused amphiphysin autoantibodies are internalized in motor neurons in vivo, where it disrupts endocytosis, leading to reduced inhibition of GABAergic interneurons causing hyperexcitability of motor neurons and SPSD-like symptoms in rats [230]. More recently, the same research group reported anxiety in mice, dysfunctional presynaptic vesicle dynamics and more as attributed effects of amphiphysin autoantibodies [231,232]. While these findings demonstrate pathological potential of amphiphysin autoantibodies, no conclusions can be drawn without validation of data.

#### 3.2.2. Drebrin Autoantibodies

Drebrin autoantibodies were first identified in four patients with adult-onset epilepsy in 2020 [233]. The antigen, drebrin, binds actin and forms stable actin filaments (F-actin) in dendritic spines, which have a pivotal role in supporting cell motility and spine shape changes through polymerization of the actin filaments [234,235]. A study examined the pathogenic effect of purified Debrin autoantibodies in primary murine hippocampal neurons [233]. Application of patient-derived drebrin autoantibodies impaired postsynaptic drebrin abundance and distribution and increased synaptic connectivity and synaptic excitability of neurons leading to neuronal network hyperexcitability. The electrophysiological effect of the drebrin autoantibodies were acute, suggesting that drebrin autoantibodies are directly pathogenic through binding to the antigen [233]. While these findings are of great interest, it is only a single publication yet, and more studies are needed to validate the pathogenicity of Drebrin autoantibodies and explore how the autoantibodies might gain access to the antigen.

## 4. Discussion

The field of AE research is rapidly expanding. New autoantibodies are discovered each year, often with distinct suspected autoantibody-mediated pathogenicity causing a wide range of clinical symptoms. The initial pathological mechanisms of the most common types are now well studied, but more rare subtypes are yet to be examined thoroughly. An exception is IgLON5 disease, possibly because it presents an intriguing link between more common neurodegenerative diseases and AE. Another exception is GAD65 autoantibodies which have been extensively studied but where the effect of autoantibody binding is still unknown.

For many of the least characterized types of AE, most of the knowledge we have are from only a single research group examining the autoantibodies of a few patients. Often, when more than one research group has investigated an autoantibody type, their results are contradictory. GABA_B_R autoantibodies are one example, where only one out of two publications could observe reduced neuronal excitability as a consequence of autoantibody binding [50,153]. Another example is CASPR2 autoantibodies that either interfere with CASPR2 cluster formation [54], increase CASPR2 cell surface expression [55], does not result in any changes in CASPR2 expression [53] or decrease synaptic CASPR2 through internalization [57,130]. These contradicting results highlight the need for, and advantages of collaborations and consistency of experimental studies on disease mechanisms between research groups. The conflicting data could perhaps be explained by differences in autoantibody binding patterns, as studies on monoclonal AE autoantibodies have revealed distinct mechanisms depending on autoantibody epitope, and the subclass of IgG autoantibodies has been shown to mediate different effector functions. This emphasizes the importance of using well-characterized autoantibodies with known IgG subtype composition from different patients to fully elucidate disease mechanisms. The use of monoclonal autoantibodies is a useful tool to distinguish which molecular and cellular aberrations are sequential and which are parallel effects of autoantibodies with different epitope targets.

For many of the AE subtypes, passive transfer in vivo disease models have been utilized to determine pathogenicity of autoantibodies. When conducting such animal experiments to examine clinical relevance of human autoantibodies, there are several limitations and potential confounding factors to consider. First, when using passive transfer models in wild type animals instead of active immunization, the introduction of a foreign peptide is highly likely to be recognized as none-self, triggering an immune response. This response may lead to symptoms beyond those specifically caused by the injected autoantibody. Second, while the target antigen of human autoantibodies is often highly homologous between rodents and humans, it is not completely identical or expressed to the same degree in the same anatomical region in both species. This disparity may lead to altered binding specificity and differences in presenting phenotype. Third, in passive transfer models, there is a lack of normal interaction between the specific autoantibody and the rest of the immune system, as observed in patients. This absence of interaction likely influences the disease phenotype. Although autoantibodies are thought to be directly disease causing, they alone cannot fully explain the entire disease phenotype in AE patients. This limitation is likely due to a complex interplay of the entire immune system, a dynamic which can not be authentically reproduced in passive transfer in vivo models.

The vast majority of studies on AE mechanisms are focused on autoantibody effects on neurons, but, many antigens of AE autoantibodies are also expressed by glial cells. An example of this is the NMDAR which is expressed by neurons, oligodendrocytes and astrocytes [236]. A single publication has studied the effects of NMDAR autoantibodies on oligodendrocytes, revealing a reduction in GLUT1, an essential protein for oligodendrocyte energy homeostasis [38]. But how does NMDAR autoantibodies affect astrocytes? The role of NMDARs in astrocytes are not well known and the question remains unanswered, but it is not unlikely that NMDAR autoantibodies could have an adverse effect on astrocyte function.

Another often unexplored aspect of AE is the cascading effects on intracellular downstream pathways. For most types of AE, only the immediate autoantibody-binding effects on the antigens has been explored, but how this affects cellular processes is yet to be clarified. Mitochondria for example change shape to compensate for the energetic needs of the cell, and blocking of AMPARs have been reported to increase mitochondrial length [237]. AMPAR autoantibodies decrease AMPAR signaling and could therefore potentially distort mitochondrial dynamics and neuronal energy homeostasis. Overall, it is essential to unravel downstream intracellular effects of autoantibody binding to be able to fully understand disease mechanisms. This would be advantageous in the understanding of clinical phenotypes and additionally expand possibilities to develop more specific targeted treatment regimens to the benefit of patients.

The majority of autoantibodies with direct pathogenic potential are directed against extracellular antigens, but a few types of autoantibodies seem to be able to interact with their antigen despite an intracellular expression. These are primarily synaptic proteins, which may be revealed to autoantibodies during exocytosis or endocytosis, during neurotransmitter release and recycling of vesicles. The process of how this occurs is not well understood.

## 5. Conclusions

The pathogenicity of the most common subtypes of AE has been established, but the pathogenic effects of the majority of subtypes have not been studied thoroughly by several independent research groups. AE research have come a long way, and is now at the border between early exploratory studies and detailed large-scale collaborative studies. The former has been successful at demonstrating direct pathogenicity of autoantibodies and has shown us the diversity of disease mechanisms between AE subtypes. Further research should now address more complex mechanistic questions and link the molecular, cellular and clinical phenotypes of AE.

## Figures and Tables

**Figure 1 cells-13-00015-f001:**
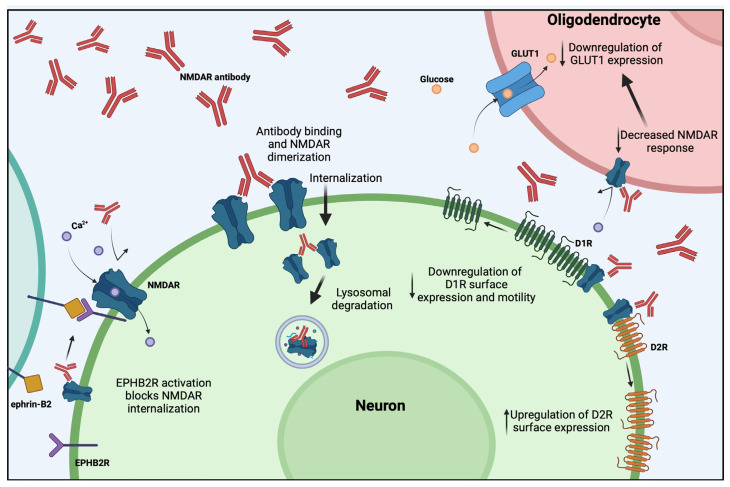
**The effect of NMDAR autoantibodies on surface receptor expression.** Autoantibodies cross-link multiple NMDARs causing internalization whereafter the receptors are degraded in lysosomes. The loss of surface NMDAR leads to other changes such as alterations in dopamine receptor surface expression in neurons and GLUT1 surface expression in oligodendrocytes. Activation of EPHB2R blocks autoantibody mediated internalization of NMDAR. NMDAR: N-methyl-D-aspartate receptor, D1R: dopamine 1 receptor, D2R: dopamine 2 receptor, GLUT1: glucose transporter 1, EPHB2R: ephrin B2 receptor. Figure created with BioRender.com (accessed on 13 December 2023).

**Figure 2 cells-13-00015-f002:**
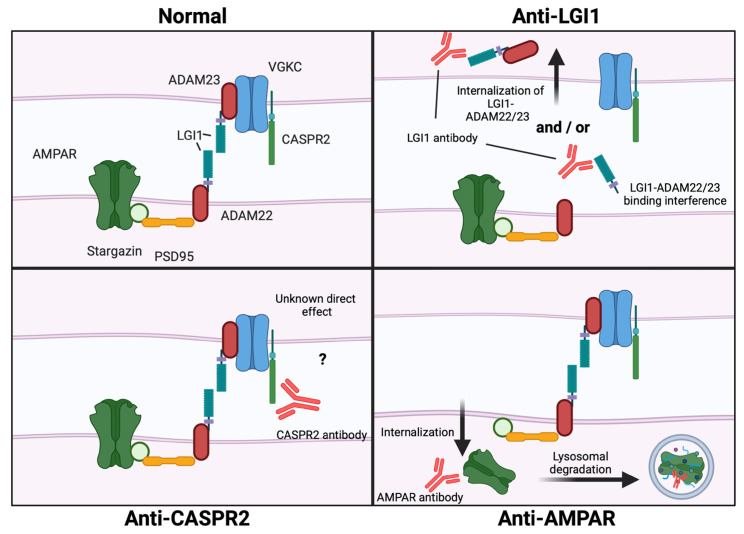
**The effect of LGI1, CASPR2 and AMPAR autoantibodies on surface expression.** All three autoantibody targets are part of the same receptor complex (**top left**). LGI1 autoantibodies (**top right**) are thought to interfere between LGI1 and ADAM22 or ADAM23, but internalization of LGI1-ADAM22/23 complexes have also been observed. The immediate effect of CASPR2 autoantibodies (**bottom left**) is still unknown. AMPAR autoantibodies (**bottom right**) cause internalization of AMPARs, which then undergo lysosomal degradation. LGI1: Leucine rich Glioma-Inactivated 1, CASPR2: Contactin-associated protein-like 2, AMPAR: α-amino-3-hydroxy-5-methyl-4-isoxazolepropionic acid receptor; ADAM22/23: Disintegrin and metalloproteinase domain-containing protein 22/23, PSD95: postsynaptic density protein 95. Figure created with BioRender.com (accessed on 13 December 2023).

**Figure 3 cells-13-00015-f003:**
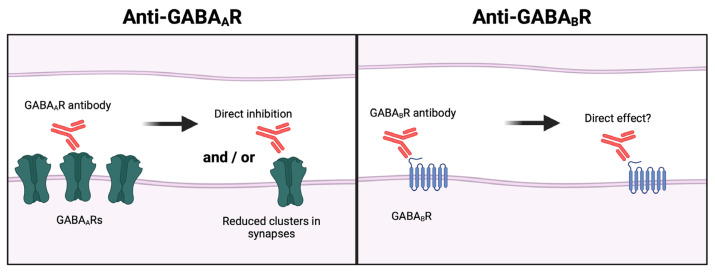
**The effect of GABA_A_R and GABA_B_R autoantibodies on surface expression.** GABA_A_R autoantibodies (**left**) either directly inhibits or reduce surface expression of GABA_A_Rs. GABA_B_R autoantibodies (**right**) either directly inhibits or cause internalization of GABA_A_Rs. Figure created with BioRender.com (accessed on 13 December 2023).

**Figure 4 cells-13-00015-f004:**
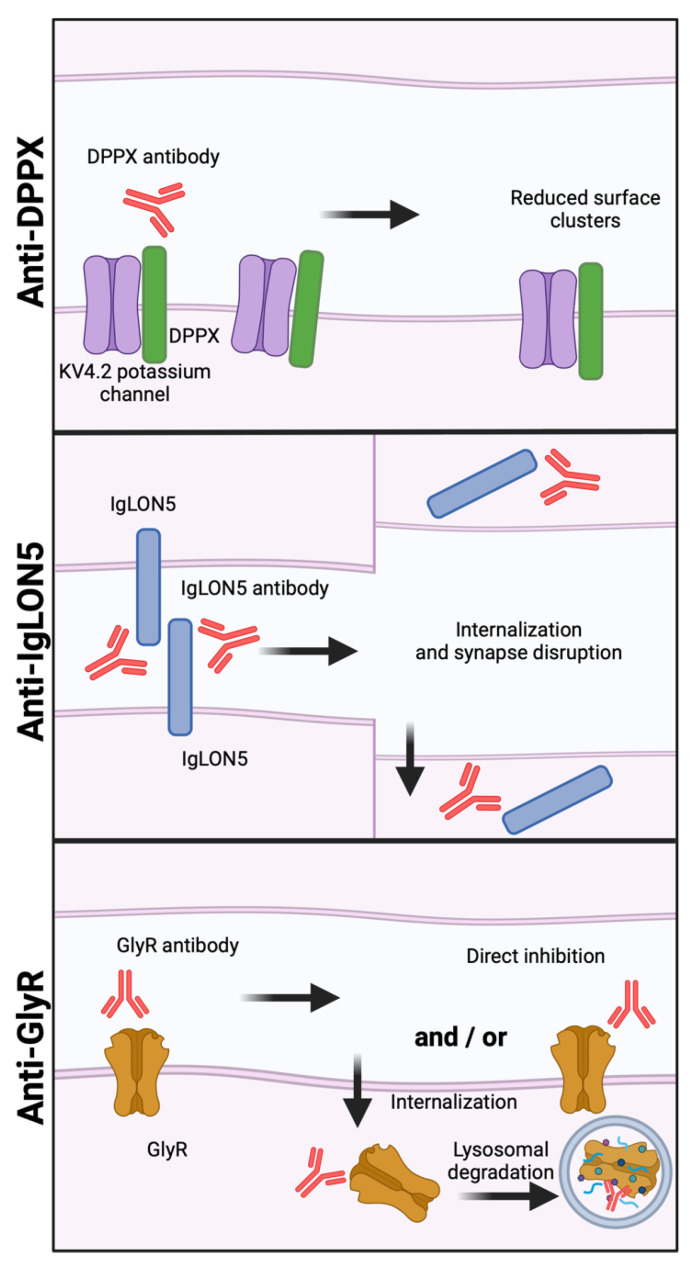
**The effect of DPPX, IgLON5, and GlyR autoantibodies on surface expression.** DPPX autoantibodies (**top**) cause a reduction in DPPX surface expression. IgLON5 autoantibodies (**middle**) internalize IgLON5 which disrupts synapses. GlyR autoantibodies (**bottom**) directly inhibits GlyR and/or cause internalization of GlyR and subsequent lysosomal degradation. Figure created with BioRender.com (accessed on 13 December 2023).

**Table 1 cells-13-00015-t001:** Classification of AE autoantibodies based on evidence level of pathogenicity.

Classification of AE Autoantibodies
**Class I**—Confirmed pathogenicity
NMDAR, LGI1, CASPR2, AMPAR, GABA_A_R, DPPX, IgLON5, GlyR, GAD65
**Class II**—Highly suspected pathogenicity
GABA_B_R, mGluR1, mGluR5, Neurexin-3α, GluK2, Ca_V_α2δ, Amphiphysin, Drebrin
**Class III**—Pathogenicity not established
D2R, mGluR2, SEZ6L2, GluRδ2

**Table 2 cells-13-00015-t002:** Overview of neuronal extracellular autoimmune encephalitides autoantibody effects.

Antigen/ IgG Subtype	Effect on the Antigen	Cellular Consequences	In Vivo Manifestations	Clinical Manifestations
**Class I: Autoantibodies with confirmed pathogenicity**				
NMDAR/ Primarily IgG1	Cross-linking, internalization and lysosomal degradation of NMDA receptors (reversible) [31,32,33,34]. Prevented by activation of EPHB2R [34,35].	Decreased neuronal excitability [36] and suppressed NMDAR dependent long term plasticity but not short term [35,37]. Decreased GLUT1 expression in oligodendrocytes [38]. Altered D1R and D2R surface expression and dynamics [39,40].	Memory impairment and depression [41,42,43]. One study reports infiltration of immune cells [44] while one reports no infiltration [43]. Seizures [45].	Psychiatric and cognitive dysfunction including anxiety and memory impairment. Dyskinesias, seizures, reduced verbal output, insomnia, autonomic dysfunction [31].
LGI1/ Primarily IgG4	Interference of LGI1—ADAM22/ADAM23 interaction [46,47,48]. One study reports LGI1—ADAM22/ADAM23 complex internalization dependent on autoantibody domain binding characteristics [49].	Increased neuronal excitability (including increased spike frequency and amplitude) [47,48,50]. Reduced synaptic AMPARs and Kv1.1 clusters [46,47]. Impaired LTP [47,49].	Memory impairments [47,49] and tight junction breakdown in BBB [51].	Limbic encephalitis, Facio-brachial dystonic seizures (FBDS) [52].
Caspr2/ Primarily IgG4	Interference of Caspr2—contactin-2 interaction [53], Caspr2 cluster formation [54], and potentially increased Caspr2 expression on the cell surface [55]. One publication reports no changes in Caspr2 surface expression [53].	Increased Kv1_2 channel expression [54,56] and Increased neuronal excitability [50,56]. Increased microglial density, altered glial morphology and raised complement C3 expression [55].	Increased sensitivity towards pain [56], memory impairments and behavioral changes [55,57].	Peripheral neuropathy, neuropathic pain, neuromyotonia, Morvan syndrome, limbic encephalitis [52].
AMPAR/ Unknown	Decrease in GluA1 and GluA2 containing AMPARS by internalization and degradation [58,59,60].	Decreased AMPAR mediated currents and impaired LTP [59,60,61].	Memory deficits and increased anxiety [60].	Limbic encephalitis, encephalopathy, seizures, prominent psychiatric symptoms [62].
GABA_A_R/ Primarily IgG1	Reduction in GABAaR clusters in synapses [63,64,65] and/or direct inhibition [66].	Decreased mean amplitude of inhibitory postsynaptic currents [64,67,68].	Seizures [68].	Subacute encephalopathy, seizures, status epilepticus [63].
DPPX/ IgG1 and IgG 4	Reduction in DPPX clusters and Kv4.2 subtype potassium channels [69,70].	Increased excitability and action potential rate of enteric neurons [70].		Encephalopathy with diarrhea and weight loss, myoclonus, seizures [71].
IgLON5/ IgG4 > IgG1	Irreversible internalization of IgLON5 clusters [72,73,74].	Decreased number of synapses, and spike rate [73,74]. Axonal degenerative changes [75] Accumulation of phosphorylated Tau and increased cell death [73].	Cognitive abnormalities, astrocyte and microglia recruitment [74]. Tau deposition [76].	Non-REM parasomnia, obstructive sleep apnea, bulbar symptoms, cognitive decline, gait instability, dystonia, chorea. May resemble neurodegenerative diseases or motor neuron disease [77].
GlycineR IgG1 and IgG 3	Internalization and lysosomal degradation of glycine receptors, and possible complement activation through binding of C3 [78]. Or, direct inhibition of GlycineR [79].	Reduced glycinergic transmission [79].	Increased anxiety [80].	Progressive Encephalopathy with Rigidity and Myoclonus (PERM), limb paralysis, cognitive impairment, seizures [78].
**Class II: Autoantibodies with highly suspected pathogenicity**				
GABA_B_R/ Primarily IgG1	No change in GABAbR expression, but direct inhibition of receptor function [81].	One study reports decreased excitability of neurons [81] while another reports no changes [50].		Limbic encephalitis, seizures [82].
mGluR1/ Primarily IgG1	Decreased number of mGluR1 clusters [83].	Reduced LTD [84].	Reversible ataxia [85]. Decreased compensatory eye movements [83].	Subacute cerebellar ataxia syndrome [83].
mGluR5/ Primarily IgG1	Decrease in mGluR5 clusters, reversible upon removal [86].			Encephalopathy, Ophelia syndrome (Hodgkins lymphoma) [86].
Neurexin-3α/ Primarily IgG1	Reduction in Neurexin-3α clusters [87].	Decreased number of synapses [87].		Confusion, seizures and decreased level of consciousness [87].
GluK-2/ Primarily IgG1	Reversible internalization [88].	Reduced GluK-2 mediated currents [88].		Headache, nausea, vomiting, decreased level of consciousness. Or acute cerebellitis with hydrocephalus [88].
Ca_V_α2δ/ Unknown		Reduced excitatory and inhibitory signaling [89].		Memory loss, psychosis and seizures [89].
**Class III: Autoantibodies without established pathogenicity**				
D2R/ Unknown				Lethargy, dystonia, agitation, confusion parkinsonism [90].
mGluR2/ Primarily IgG1				Cerebellar ataxia [91].
SEZ6L2/ Primarily IgG4				Dysarthria, ataxia, bradykinesia [92].
GluRδ2/ Unknown				Cerebellar ataxia [93].

## Data Availability

Not applicable.

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
