# Peer review of "Pathophysiological Effects of Autoantibodies in Autoimmune Encephalitides"

_cells, 2023, doi:10.3390/cells13010015_

Round 1

Reviewer 1 Report

Comments and Suggestions for Authors

In this manuscript, Ryding et al. review the current literature on autoantibodies targeting both extracellular and intracellular antigens associated with autoimmune encephalitis. They present evaluation criteria that could be used to define the pathogenicity of the antibodies and they summarize currently known pathogenic mechanisms.

I have the following comments/questions:

Line 45: I would rephrase saying “Antibodies are one of the link”, since chemokines, cytokines and other direct cell-cell contact also link the adaptive and innate immune system.

Line 68: I would rephrase for clarity. For example: “In AE, the pathogenic roles of several autoantibodies have...”

- The formatting of table 2 left to be desired. It is too compressed, and it would also help to have vertical lines to facilitate reading the text.

Line 439: I would rephrase this sentence. It is the effects of the binding of IgLON5 autoantibodies that are irreversible, not the binding itself which is usually quite strong.

Line 619: I would rephrase to say that the antibodies can have direct access to the antigens, since during endocytosis, the antigens are not exposed to the extracellular space, but the antibodies are internalized and encounter their antigens only after fusion with other vesicles or organelles. In the referred paper, the antibodies are internalized during endocytosis, but the antigens were brought on the surface of the cell by exocytosis (only the first contact hypothesis). The authors should provide a reference for the other contact hypothesis if available.   

I believe the authors should discuss about confounding factors inherent to animal experimentations aimed at confirming the clinical relevance of autoantibodies. Antibody subclasses and FcR are similar, but not identical between rodents and humans. The authors could add to the manuscript a small section discussing the variations and why this could impact the phenotypes observed during in vivo experimentations, especially if the model relies on infusion of human antibodies in WT rodents. Also, the authors could discuss why the injection of a foreign peptide could trigger the immune system of the rodents independently of the antigen-specificity or -pathogenicity. Did the researchers use humanized rodents in the cited manuscripts, were they using other genetically modified strains? etc.  

I found the following typos while reading the manuscript.

Line 65: I suppose the authors mean “triggered”

Line 79: replace “have” by “has”

Line 317: replace “has” by “have”

Line 357: “mouse”

Line 358: “Others have reported that”

Line 438: “after the passive”

Line 441: “mice”

Line 506: “mouse”

Line 535: missing comma after 2019

Line 545: “important”

Line 615: either write “intracellularly located” or remove “located”

Line 618: “mechanism explaining how”

Line 743: please verify the sentence.

Line 747: “result in any changes in “

Line 748: “highlight”, also please rephrase the sentence (line 748-749)

Line 754: “well-characterized”

Line 760: remove “both”

Line 763: “affect”

Line 769: “processes”

Final thoughts: The authors have submitted a well-written manuscript that summarize the current knowledge we have on the subject. If the authors can address my comments, I believe the manuscript can be accepted for publication.  

Comments on the Quality of English Language

See comments above

Author Response

Thank you for taking the time to review our manuscript, and for all of your comments and corrections. It has undoubtedly improved the quality of our paper.

Please find our point-to-point corrections below:

Line 45: I would rephrase saying “Antibodies are one of the link”, since chemokines, cytokines and other direct cell-cell contact also link the adaptive and innate immune system.

Response: We have rephrased the sentence so that it now is more correct.

Line 68: I would rephrase for clarity. For example: “In AE, the pathogenic roles of several autoantibodies have...”

Response: We have rephrased the sentence.

- The formatting of table 2 left to be desired. It is too compressed, and it would also help to have vertical lines to facilitate reading the text.

Response: We agree. We have changed the table back to its original format, however, the editors of Cells may not accept this as it might differ too much from their standard format.

Line 439: I would rephrase this sentence. It is the effects of the binding of IgLON5 autoantibodies that are irreversible, not the binding itself which is usually quite strong.

Response: Thank you for your comment. We have rephrased it and hope that it is more easily to understand now.

Line 619: I would rephrase to say that the antibodies can have direct access to the antigens, since during endocytosis, the antigens are not exposed to the extracellular space, but the antibodies are internalized and encounter their antigens only after fusion with other vesicles or organelles. In the referred paper, the antibodies are internalized during endocytosis, but the antigens were brought on the surface of the cell by exocytosis (only the first contact hypothesis). The authors should provide a reference for the other contact hypothesis if available. 

Response: We have changed the section so that the differences between the two hypotheses are more clear. We have also changed the reference to one that describes both hypotheses. (line 655).

I believe the authors should discuss about confounding factors inherent to animal experimentations aimed at confirming the clinical relevance of autoantibodies. Antibody subclasses and FcR are similar, but not identical between rodents and humans. The authors could add to the manuscript a small section discussing the variations and why this could impact the phenotypes observed during in vivo experimentations, especially if the model relies on infusion of human antibodies in WT rodents. Also, the authors could discuss why the injection of a foreign peptide could trigger the immune system of the rodents independently of the antigen-specificity or -pathogenicity. Did the researchers use humanized rodents in the cited manuscripts, were they using other genetically modified strains? etc.

Response:  Thank you for this very relevant comment. We have now addressed some of the limitations in the discussion (line 817). A single paper referred to in this review used relevant humanized animals (line 475), although the added human gene is not related to the immune system, but the specific disease phenotype (human Tau protein). Almost all in vivo studies used passive transfer models.

I found the following typos while reading the manuscript.

Response: Thank you for taking the time to find, and inform us of these mistakes. Especially “Line 743: please verify the sentence“ which allowed us to find a citation error and a misunderstanding in the GABABR section.

Reviewer 2 Report

Comments and Suggestions for Authors

I have analysed the manuscript of Ryding et al, submitted to Cells, and I find it remarkable. It is exploring all the field of Autoimmune encephalitides in a comprehensive way, it reads well, it is well documented and easy to understand for the readers. I recommend acceptance of the manuscript. However, since it is well written and practical, I would like to see a comment about the issue of the cutt-off of antibodies titer. It is worthy to answer to the following questions:

- do we produce antibodies to such antigens, normally, in low amounts?

- at what titer of abs we decide that the autoimmune mechanism is responsible for a neurological syndrome?

- how is established in practice this cutt-off, might it be a source of medical errors?

Thank you,

Prof. Bogdan Ovidiu Popescu

Carol Davila University Bucharest, Romania

Author Response

Thank you for taking the time to review our paper, your uplifting comments, and your suggestion in regards to antibody titers. We have added a section (line 75) informing of the presence of serum neuronal autoantibodies in the healthy population, and what diagnostic strategies are recommended to avoid false positive results.

Reviewer 3 Report

Comments and Suggestions for Authors

In this manuscript, the authors undertake a comprehensive exploration of the current understanding pertaining to the molecular and cellular implications induced by autoantibodies associated with autoimmune encephalitis. Furthermore, a meticulous assessment is conducted to scrutinize the substantiating evidence supporting the suggested pathophysiological mechanisms through which autoantibodies manifest their impact in autoimmune encephalitis. The review explores the interplay between these autoantibodies and their specific molecular and cellular targets, contributing to an enhanced comprehension of the multifaceted dynamics characterizing the pathogenesis of autoimmune encephalitis.

The paper is well thought-out, featuring clear explanations and accompanied by illustrative figures. However, there are a few typos and grammar errors that need double-checking. Despite these minor issues, the overall quality of the paper is commendable, making it an engaging and insightful read for the interested readers.

Comments on the Quality of English Language

Minor grammar and spelling mistakes

Author Response

Thank you for taking the time to review our paper and your uplifting comments. We have taken the time to carefully read the manuscript again and correct all grammar and spelling mistakes.